

# The effect of drop-in centers on access to HIV testing, case finding, and condom use among female sex workers in Addis Ababa, Ethiopia

Saro Abdella Abrahim[1], Meaza Demissie[2], Alemayehu Worku[3], Merga Dheresa[4] and Yemane Berhane[5]

[1] HIV and TB Research Directorate, Ethiopian Public Health Institute, Addis Ababa, Ethiopia
[2] Department of Global Health and Health Policy, Addis Continental Institute of Public Health, Addis Ababa, Ethiopia
[3] School of Public Health, College of Health Sciences, Addis Ababa University, Addis Ababa, Ethiopia
[4] School of Nursing and Midwifery, College of Health Sciences, Haramaya University, Harar, Ethiopia
[5] Department of Epidemiology and Biostatistics, Addis Continental Institute of Public Health, Addis Ababa, Ethiopia

## ABSTRACT

**Background:** Varied HIV prevention interventions involving multiple strategies has been instrumental in the effort to contain and lessen the prevalence of HIV around the globe. However, female sex workers (FSWs) often face stigma and discriminatory challenges, resulting in lower access to the HIV prevention initiatives. This study has aimed to assess the effect of one of the HIV service delivery models, the Drop-in Centers (DICs), which is designed to overcome the service uptake barriers of FSWs.
**Method:** A quasi-experimental study design was employed. A respondent-driven sampling technique was used to recruit 1,366 FSWs from January to June 2020. A propensity score matching technique was used to balance the potential confounders between FSWs who had access to DICs and those who had never accessed DICs. Comparisons of the effect of DIC on the outcome of interest was made using a logit regression model at a 5% level of significance.
**Results:** A total of 1,366 FSWs took part in the study. The analysis estimated the average treatment effects of access to DICs on four key outcomes: ever-testing to know HIV status, finding HIV-positive FSWs, awareness of HIV-positive status, and consistent condom use. A significant effect of DIC was seen at a 95% confidence interval on each outcome. Access to DIC produced a 7.58% increase in the probability of testing to know HIV status ($P < 0.001$), a 7.02% increment in finding HIV-positive FSWs ($P = 0.003$), an increase of 6.93% in awareness of HIV status among HIV positive FSWs ($P = 0.001$), and a 4.39% rise in consistent condom use ($P = 0.01$).
**Conclusions:** Ensuring access of FSWs to DICs has led to an upsurge in HIV testing among FSWs, raising HIV status awareness among those who are HIV positive, and encouraged consistent condom use. To provide effective HIV prevention services, particularly to those FSWs living with HIV, it is essential to strengthen the services provided in DICs and expand the centers. This will ensure that the entire network of FSWs is reached with appropriate HIV prevention services.

Corresponding author
Saro Abdella Abrahim,
helen_saro@yahoo.com

# INTRODUCTION

Female sex workers (FSWs), gay men, prisoners, those who inject drugs, and transgender individuals all face social and legal issues due to their behaviors (*UNAIDS, 2020*). In 2020, evidence has shown that these key populations and their clients comprise two-thirds of global HIV infections (*UNAIDS, 2021b*). In particular, FSWs have a thirtyfold greater risk of contracting HIV than their counterpart in the general population in which they reside (*UNAIDS, 2021a*). Given these alarming statistics, a combination of prevention interventions and strategies has been designed (*Tao et al., 2018*). This includes improved education on HIV/AIDS prevention; early diagnosis of people with HIV; timely treatment with antiretroviral drugs; the provision of condoms; expanding access to clean needles for drug users; and provision of pre-exposure prophylaxis for those who are not infected with HIV (*Kabapy, Shatat & Abd El-Wahab, 2020*). However, FSWs frequently experience difficulties accessing HIV prevention initiatives, primarily due to the stigma and discrimination they encounter (*Nyato et al., 2019*). Additionally, they have limited awareness of the availability and effectiveness of these interventions. Gender-based violence also contributes to their increased vulnerability to HIV (*Tadesse et al., 2020*). Furthermore, there is limited knowledge among programme planners and HIV experts on effective approaches for reaching out to FSWs (*Scorgie et al., 2012*; *Shannon et al., 2009*). All these factors contribute to the low access to prevention interventions for FSWs. Ultimately, new HIV infections were also observed more among FSWs than women in the general population globally (*Joint United Nations Programme on HIV/AIDS, 2021*).

To reduce the higher rate of HIV infection in FSWs, it is essential to avail comprehensive health services and testing (*Chersich et al., 2012*). Nonetheless, the availability of the services does not guarantee equal utilization by FSWs and other population groups (*Nyato et al., 2019*). Hence, initiatives with a different modality of service delivery that account for the challenges FSWs face while attempting to visit health facilities are required to maximize the benefits of using these services (*Shantanam & Mueller, 2018*). Drop-In Centers (DICs) offer a unique and convenient solution for those at risk of HIV/AIDS to access vital prevention services. These centers provide a safe and supportive environment for FSWs to access resources without judgment or fear (*Suraksha*). The innovative nature of these centers makes them an important part of the effort to reduce HIV transmission rates by delivering essential prevention services. It also provides opportunities for FSWs to stand collectively against violence or other forms of discrimination affecting sex workers (*UNAIDS, 2012*). Sex worker-led initiatives that advocate for policy changes to improve the rights of sex workers are also essential in tackling HIV amongst this vulnerable population (*Vu et al., 2020*). The World Health Organization (WHO) recommended that countries consider establishing safe spaces or drop-in centers to seek HIV prevention services for at-risk populations without fear of stigma or judgment (*World Health Organization, 2012*).

As a result, DICs became an important part of HIV prevention and care for FSWs. They provide crucial services such as HIV testing, counselling, education, health promotion materials and activities, medication assistance, outreach to HIV-positive people, employment support, and referral services. These centers also offer condoms and safer sex education messages to encourage the development of risk-reduction behaviors among FSWs. With the diverse population at a DIC, there is an immense potential for building a community of people living with HIV/AIDS (PLHIV) to support each other (*Suraksha*; *Taghizadeh et al., 2015*; *Valerius et al., 2015*). Studies have shown that establishing and implementing DICs can be helpful in effectively linking marginalized FSWs with sexual and reproductive health services (*Suraksha*; *Kim et al., 2015*).

In Ethiopia, about one in every five FSWs is HIV positive (*Abdella et al., 2022*), access to HIV prevention services is limited, and some face stigma and discrimination (*Federal HIV/ AIDS Prevention and Control Office, 2018*). Recognizing these and aiming to improve access to high-impact HIV prevention interventions targeting FSWs, DICs were established in Addis Ababa and major towns in the Amhara region with the support obtained from PEPFAR through USAID in 2016. Although the DICs have been provided comprehensive HIV prevention, care, and treatment services for 4 years, their effect on increasing access to prevention interventions has never been assessed. By evaluating the effect of these DICs, it is possible to gain insights into how they have improved the lives of FSWs who access them. This information can then be used to identify areas for improvement or expansion. Additionally, assessing the effectiveness of DICs will also inform funding decisions.

## MATERIALS AND METHODS

The quasi-experimental study design compared HIV prevention interventions in FSWs who have previously accessed DICs (ADICs) and those who have never accessed DICs (NDICs). Not all FSWs have previously accessed DICs; some FSWs even do not know the existence of such a modality. Those FSWs who ever get any service from the center are categorized as have previously accessed DICs (ADICs), and those who have never used DIC are grouped as never have accessed DICs (NDICs). Self-reported histories of access to these centers were used to assign participants to either of the groups.

### Study setting and period

The study was conducted in Addis Ababa, the capital city of Ethiopia, from January to June 2020. Two temporary data collection sites were established in the city. The sites were purposefully selected to get as many FSWs as possible in the city near establishments like bars, hotels and groceries. The research team rented houses for 6 months to serve as data and sample collection sites.

### Study population sampling procedure

All FSWs residing in Addis Ababa who have previously accessed DICs and who have never accessed were targeted to be enrolled in the study using a respondent-driven sampling technique, a special form of the chain referral sampling method (*Heckathorn, 1997*). A

total of 12 seeds were purposefully selected to recruit study participants. The number and types of seeds were determined based on a formative assessment conducted before the survey. When selecting FSWs, various criteria were considered, such as the modality of meeting their clients (bars and/or hotels, red-light houses, local drinking houses, on streets, and *via* hidden cell phone services), age (<25, 25–30, >30), and their residence locations in the city. The number of seeds selected for the city was also based on the estimated total number of FSWs living in Addis Ababa, obtained from the formative assessment. Each seed was given three coupons to invite FSWs to form the first wave. The recruitment was continued until the intended sample size was reached. Details of the method are available elsewhere (*Abdella et al., 2022*).

All females aged 15 years and older who received money or other benefits in exchange for sex in the last 30 days, lived in the city for at least 1 month, provided consent for participation, and possessed a valid coupon were eligible to participate in the study.

## Sample size determination

Participants were classified based on their exposure status as exposed (have previously accessed DICs) or non-exposed (have never accessed DICs). The double population proportion formula was used to determine a minimum sample size using the following formula and assumption:

$$n = D \frac{[Z_{1-\alpha}\sqrt{2P(1-P)} + Z_{1-\beta}\sqrt{P_1(1-P_1) + P_2(1-P_1)}]^2}{(P_2 - P_1)^2}$$

where z is the upper $\alpha/2$ point of the standard normal distribution with $\alpha = 0.05$ significance level and a *P*-value of 46%, because no study reports the HIV test uptake proportion in FSWs in Ethiopia, a pooled HIV test uptake proportion in 14 East African countries was used (*Mulholland et al., 2022*). The average intervention effect to enhance HIV testing in FSWs in Kenya was 10% (*Kelvin et al., 2019*).

To achieve a power of 80% for detecting a difference in proportions of 0.10 between the two groups (test and reference group) at a two-sided *P*-value of 0.05, and after applying continuity correction, we reached the required sample size of 409 for each group (*i.e.*, a total sample size of 818, assuming equal group sizes).

## Dependent and independent variables

Dependent variables were HIV prevention interventions, including life-time HIV tests, finding HIV-positive FSWs, being aware of their HIV-positive status, and consistent condom use.

A lifetime HIV test was measured as a self-reported previous HIV test status. Participants were asked whether they have ever tested for HIV before; those who reported previous tests and received results were recorded as having received lifetime HIV tests.

Finding HIV positive FSWs: We considered all FSWs with a current HIV-positive test result to constitute a group of HIV-positive FSWs. Rapid HIV tests were conducted on every participant during the study period.

Aware of their HIV-positive status: During the survey, FSWs who tested positive for HIV were asked if they knew their current result before or not. Those who said they knew they had the virus were seen as aware of their HIV-positive status.

Self-reported consistent condom use: Using a condom in every sexual act in the past 30 days as reported by the study participants.

Age, education, marital status, monthly income, drinking alcohol, and access to DICs were independent variables.

## Methods of data collection

Trained interviewers, laboratory technologists, and data managers were assigned to collect data using a structured questionnaire. The interview tool was designed and uploaded to an open-source data collection tool known as ODK (Open Data Kit). This platform was configured to prevent any missed data; subsequent questions would not be opened unless the preceding one was answered. Findings from a formative assessment conducted before this study were used to design the language and order of the questions in the questionnaire. Each question was written and not leading so as not to bias participant responses. Finally, the questionnaire was tested at Bishoftu town, near the study city. Feedback from the testing was used as input to revise the questionnaire. All the collected information from the sites was automatically deposited daily into the Ethiopian Public Health Institute server.

Blood was collected to conduct rapid HIV tests at the study sites using the National HIV Testing Algorithm (HIV1/2 stat-pak, assay one; Abone, assay two; and SD bioline, assay three), and results were returned to the participants. Quality assurance activities were in place during testing (Abdella et al., 2022).

## Methods of data analysis

Participants were categorized as those who have previously accessed DICs and who have never accessed DICs. The two groups were matched for potential co-variates (age, education, marital status, monthly income, and alcohol usage) using the propensity score matching technique. A Z-test was used to see the differences in the proportion of outcome variables between the two groups. ATEA was used to determine the effect of access to DICs on each outcome variable.

We used Stata V14.1 software to test four hypotheses: (1) the effect of ADICs on lifetime HIV test; (2) the effect of ADICs on finding HIV positive; (3) the effect of ADICs on awareness of their HIV-positive status, and (4) the effect of ADICs on consistent condom use. We have used a logit regression model to analyze the effect of ADICs on the outcome variables at a 5% level of significance.

An outcome variable with a positive sign and a significant probability coefficient at a P-value of less than 0.05 is considered an effective outcome variable. The level of effectiveness is quantified based on the magnitude of the coefficient.

## Sensitivity analysis

The impact of unobserved confounding on the matching model was evaluated using Mantel-Haenszel bound estimates (mhbounds). MH bounds calculate Rosenbaum's
bounds for average treatment effects on the treated in the presence of unobserved heterogeneity (hidden bias) between treatment and control cases, where both treatment and response variables are binary. MH bounds can be used directly after running the necessary matching strategies (*Becker & Caliendo, 2007*). We employed the Mantel-Haenszel test to check the sensitivity of the estimated average treatment effect on DICs due to unobserved heterogeneity (hidden bias) between those who have previously accessed DICs and whose who have never accessed DICs. Under the assumption of no hidden confounders when gamma is equal to one and at a 99% confidence interval, the estimated average treatment effect of ADICs is not sensitive to hidden bias on all the outcomes: lifetime HIV test, finding HIV positives, HIV status awareness among HIV positives, and consistent condom use (Annex 1).

## Ethical considerations

Participants voluntarily opted to participate in the study and consented to be interviewed. All participants provided written informed consent to participate. Mature minors, aged between 15 and 18 years, who were responsible for their livelihood were also involved in the study after providing consents for themselves. No personal identifying information was collected, and participants were given identification numbers (IDs) instead. HIV positive FSWs identified during the study were provided with viral load test and linked to care at health facilities. Interviews and blood draws were conducted in a private room, guaranteeing the comfort and privacy of all participants. Incentives for participation and compensation for the transport cost to the study site were provided to all study participants. The Ethiopian Public Health Institute's Research Ethics Institutional Review Board reviewed and approved the study protocol for implementation (Ref. EPHI 6.13/517).

# RESULTS

## Background characteristics of participants

A total of 3,178 coupons were distributed, resulting in 1,394 FSWs showing up at the study sites. Of these, 1,366 (98%) met eligibility criteria and were enrolled in to the study. There were no dead or unproductive seeds. The minimum wave length was two, and the maximum was sixteen. Most of the study participants, 866 (63.4%), were in the age range of 20–29 years. More than half, 768 (56.2%) of the respondents were never married, and 624 (45.7%) have attended primary 2nd cycle classes (grades 5–8). Very few participants, 67 (4.9%), have a monthly income above 10,000 Ethiopian birr, 756 (55.4%) took alcohol twice or more per week, and 412 (30.2%) have previously accessed DICs (Table 1).

In the unmatched comparison, 412 FSWs have previously accessed DICs, and 945 have never accessed DICs. After matching, we kept 822 samples (412 in ADICs and 410 in NDICs) for analysis. Standardized differences were calculated for the different features between unmatched and matched groups, with imbalance being defined as an absolute value greater than 0.1 (small size effect) (*Gebrehiwot & van der Veen, 2015*). Before matching, FSWs who have previously accessed DICs and those who have never accessed DICs differed significantly with age, monthly income, and alcohol consumption

**Table 1 Socio-demographic characteristics of FSWs in Addis Ababa, 2020.**

| Sr. No | Variable | Categories | Number | Percent |
|---|---|---|---|---|
| 1 | Age in year | 16–19 | 232 | 17.0 |
| | | 20–29 | 866 | 63.4 |
| | | 30–39 | 246 | 18.0 |
| | | 40–59 | 22 | 1.6 |
| 2 | Marital status | Never married | 768 | 56.2 |
| | | Married | 55 | 4.0 |
| | | Divorced/separated/widowed | 543 | 39.8 |
| 3 | Educational status | Not education | 260 | 19.0 |
| | | Primary 1st cycle (grade 1–4) | 156 | 11.4 |
| | | Primary 2nd cycle (grade 5–8) | 624 | 45.7 |
| | | High school and above | 326 | 23.9 |
| 4 | Income per month in Ethiopian Birr | Less than 3,000 | 556 | 40.7 |
| | | 3,000–10,000 | 743 | 54.4 |
| | | Above 10,000 | 67 | 4.9 |
| 5 | Drinking alcohol | Never drink | 409 | 30.0 |
| | | Less or once per month | 69 | 5.1 |
| | | 2–4 times per month | 132 | 9.7 |
| | | More than 5 times per month | 756 | 55.3 |
| 7 | Access to DIC | Yes | 412 | 30.2 |
| | | No | 954 | 69.9 |

(standardized difference >0.1). After matching, the procedure successfully eliminated the differences between the two groups on all the covariates. Therefore, all the covariates were balanced, and the standardized differences were below 0.1 (Table 2 and Annex 2).

In the pool of balanced groups, the frequency and proportion of FSWs with the test to know HIV status, HIV positives, HIV status awareness among HIV positives, and consistent condom use were described. The number of FSWs who had ever tested to know their HIV status was 765 (93.1%), of which 75 (9.1%) received HIV-positive results. During the survey period, the number of HIV-positive FSWs was 129 (15.7%). However, only 75 (58.1%) were aware of their HIV-positive status before. The majority, 779 (94.7%), of FSWs reported consistent condom use (Table 3).

DICs had a significant effect on all the outcomes of interest at a 95% confidence interval. Access to DICs increases the probability of testing to know HIV status by 7.6% ($P < 0.001$), finding HIV-positive FSWs by 7.0% ($P = 0.003$), increasing awareness of HIV status among HIV-positive FSWs by 6.9% ($P = 0.001$) and consistent condom use by 4.4% ($P = 0.01$) (Table 4).

## DISCUSSION

Our study's results indicate that DICs positively impact HIV tests uptake and encourages consistent condom use among FSWs. Access to DICs enhances the likelihood of FSWs

**Table 2 Covariates balance among FSWs who have access to DICs and have no access to DICs in unmatched and matched cohorts, 2020.**

| Covariates | Unmatched (n = 1,366; ADICs = 412, NDICs = 954) | | | Standardized differences | Propensity score matched (n = 824; ADICs = 412, NDICs = 410) | | | Standardized differences |
|---|---|---|---|---|---|---|---|---|
| | ADICs | NDICs | t-value | | ADICs | NDICs | t-value | |
| Age of FSWs | 25.847 | 26.613 | −2.17 (P = 0.030) | −0.1316192 | 25.847 | 25.478 | 1.02 (P = 0.308) | 0.0710199 |
| Ever attend school | 0.79854 | 0.81447 | −0.69 (P = 0.492) | −0.040277 | 0.79854 | 0.77913 | 0.68 (P = 0.495) | 0.0475318 |
| Ever married | 0.42476 | 0.4434 | −0.64 (P = 0.524) | −0.0375802 | 0.42476 | 0.45388 | −0.84 (P = 0.400) | −0.0586402 |
| Monthly income >5,000 | 0.28641 | 0.39413 | −3.82 (P < 0.001) | −0.2286488 | 0.28641 | 0.27913 | 0.23 (P = 0.817) | 0.0161498 |
| Ever drink alcohol | 0.60194 | 0.74319 | −5.28 (P < 0.001) | −0.3041741 | 0.60194 | 0.62379 | −0.64 (P = 0.520) | −0.0448036 |

**Table 3 The descriptive proportion of outcome variables in propensity-matched groups of FSWs in Addis Ababa, 2020.**

| Variables | Number (%) | Number (%) | | Z-test |
|---|---|---|---|---|
| | ADICs & NDICs n = 822 | ADICs n = 412 | NDICs n = 410 | <0.001 |
| Ever tested for HIV | 765 (93.1%) | 401 (97.3%) | 364 (88.8%) | <0.001 |
| Current HIV positive | 129 (15.7%) | 83 (20.2%) | 46 (11.2%) | <0.001 |
| Aware of their HIV-positive status | 75 (58.1%) | 57 (68.7) | 18 (39.1) | <0.001 |
| Consistent condom use | 779 (93.7%) | 395 (95.9%) | 375 (91.5%) | 0.06 |

Note:
ADICs–have previously accessed drop-in centers; NDICs—have never accessed drop-in centers.

**Table 4 The estimated average effect of access to drop-in centers on outcome variables among FSWs in Addis Ababa, 2020.**

| ATET on DIC (ADIC vs. NDIC) | | Sample size | Coef. | Std. Err. | P-value | 95% CI |
|---|---|---|---|---|---|---|
| Ever tested for HIV | Yes | 1,256 | 0.0758 | 0.0187 | <0.001 | [0.039–0.112] |
| | No | 110 | | | | |
| Finding HIV-positive FSWs | Yes | 228 | 0.0702 | 0.0239 | 0.003 | [0.023–0.117] |
| | No | 1,138 | | | | |
| Aware of their HIV-positive status | Yes | 108 | 0.0693 | 0.0310 | 0.001 | [0.026–0.112] |
| | No | 1,132 | | | | |
| Consistent condom use | Yes | 1,255 | 0.0439 | 0.0171 | 0.010 | [0.010–0.078] |
| | No | 111 | | | | |

undergoing HIV testing to determine their HIV status, identifying HIV-positive FSWs, promoting awareness of HIV status among HIV-positive people, and increasing condom use. Although there has been plenty of evidence that demonstrates less access to HIV prevention interventions by FSWs due to structural, behavioral, and service-related barriers (*Kabapy, Shatat & Abd El-Wahab, 2020*; *Scorgie et al., 2012*), our findings support that DICs can be utilized to successfully link FSWs to some of the vital prevention interventions, particularly HIV testing services and condom use.

Due to the ongoing risk of HIV acquisition in FSWs, they should test for HIV regularly, preferably once yearly (*Johnson, 2015*). Regular testing allows HIV-positive FSWs to take ART as early as possible for healthy living. HIV-negative test results also encourage HIV-negative FSWs to maintain their status (*Suraksha*; *Dong et al., 2019*). Our study shows that access to DICs increases the probability of HIV testing among FSWs significantly (7.6%). This result aligns with a finding from a review study that shows particular modalities of service delivery, like DICs, improve access to sexual and reproductive health services (*Valerius et al., 2015*). Such centers provide HIV, STIs, and TB screening and education services free of charge by trained personnel (*Deering et al., 2009*) and hence address some barriers to HIV testing, including financial constraints, a low perception of acquiring HIV, and poor HIV knowledge (*Tokar et al., 2018*). Therefore, it is essential to make the testing service accessible for FSWs through expanding DICs to enable them to test for HIV at least once a year so that they get treatment immediately to attain viral suppression.

HIV testing for FSWs living with HIV is critical to break the HIV transmission cycle. Early treatment of HIV facilitates viral suppression to an undetectable limit, making it less likely for someone to transmit the virus during unprotected sex (*Cohen et al., 2016*). However, in many African countries, only a small proportion of FSWs have access to HIV testing; for instance in Benin, the HIV testing was 60.6% (*Morin et al., 2021*), and in Zimbabwe it was 64% (*Cowan et al., 2017*). The main reason for the low HIV testing in HIV positive FSWs could be that, unlike FSWs who do not know their status, HIV positive FSWs suffer dual stigma and discrimination due to their work and HIV-positive status (*Oguntibeju, 2012*; *Nnko et al., 2019*). Interestingly, our findings not only show the effect of DICs in increasing access of all FSWs to HIV testing, but also it demonstrates that DICs upsurge access to HIV testing in HIV positive FSWs significantly (6.9%). The reason could be that DICs are designed to overcome barriers that FSWs encounter when attempting to access conventional health services for prevention, care, and treatment of sexual and reproductive health, including HIV testing. The centers provide nonjudgmental, safe, and enabling environments where social and health services can be accessed (*Deering et al., 2009*). Moreover, this model of service delivery is established based on the community empowerment concept, which is defined as a "collective process through which the structural constraints to health, human rights, and well-being are addressed by sex workers to create social and behavioral changes, and access to health services" (*Nyato et al., 2019*). Hence, it is essential to maintain and strengthen the programs at DICs, particularly psychosocial support, to attract FSWs living with HIV. This would enable them to test for HIV and be put on ART to achieve viral suppression.

Our study also shows that the likelihood of finding HIV-positive FSWs in those who have previously accessed DICs is 7.0% higher than in those who have never accessed. The reason for more HIV-positive FSW cases in those who have previously accessed DICs could be that the centers are supplemented by peers, friends, and healthcare workers to provide psychological support to HIV positive FSWs and educational activities, which are essential for HIV prevention and control (*Tokar et al., 2018*). Hence, many HIV positives FSWs might have been referred to DICs from different HIV test providing outlets such as

clinics, hospitals and outreach services. Finding HIV positive population clustered in DICs offers an excellent opportunity to avail HIV care and treatment services efficiently. Therefore, increasing the number of DICs is critical and efficient to reach as many HIV-positive FSWs as possible.

In this study, consistent condom use in the past 30 days was 93.67%, which is comparable with the study results from Togo (95.6%) (*Bitty-Anderson et al., 2022*), and the Dominican Republic (95.2%) (*Carrasco et al., 2019*). Access to DICs increases the probability of consistent condom use by 4.39%. This is because FSWs with access to DICs attend counselling, group meetings, and share experiences with their peers that encourage consistent condom use (*United Nation Office on Drugs and Crime, 2015*). In addition, a free condom is also available at the centers to help FSWs always use them (*Population Services International, 2020*). As sexual intercourse contributes to more than 90% of HIV infections in Sub-Saharan Africa (*Wilson, 2002*), consistent condom use is the most effective way to prevent HIV transmission in this region (*Addis Alene, 2014*). Thus, consistent availability of condoms, education about proper condom use, and maintaining the quality of other services provided in the centers should be strengthened.

We recruited adequate samples of FSWs using the RDS technique to reach all categories of FSWs. The technique also maximizes the deeper penetration into the FSWs' network by allowing participants to select the respondents themselves, which would never be possible otherwise. The study also used propensity score matching to balance the comparison groups and reduce the potential for selection and confounder bias.

The possibility of social desirability bias is one of the limitations of this study when collecting data regarding variables like consistent condom use. To minimize such bias, the data collectors clarified to the participants that they do not judge their responses. The other limitation could be that, although the two comparison groups were matched for many variables, there may be a chance of masked confounding variables that were not considered. However, we have run a sensitivity analysis and confirmed that the effect of the masked confounders in this study is insignificant enough to bias the findings.

## CONCLUSIONS

In summary, the findings of this study show that access to DICs has positive effect on the increase in HIV testing, HIV case finding, HIV status awareness among HIV positives, and consistent condom use by FSWs. the findings of this study indicate that access to DICs has a positive effect on various aspects related to HIV prevention among FSWs. These aspects include increased HIV testing, improved identification of HIV cases, enhanced awareness of HIV status among those already infected, and a higher rate of consistent condom use among FSWs. Consistently availing HIV prevention services such as HIV testing and condom provision at DICs is critical to improve service uptake by FSWs. It is also essential to expand the sites of DICs targeting all FSWs.

## ACKNOWLEDGEMENTS

We would like to acknowledge the study participants, data collectors, and supervisors.

### Funding
The authors received no funding for this work.

### Competing Interests
The authors declare that they have no competing interests.

### Author Contributions
- Saro Abdella Abrahim conceived and designed the experiments, performed the experiments, analyzed the data, prepared figures and/or tables, authored or reviewed drafts of the article, and approved the final draft.
- Meaza Demissie conceived and designed the experiments, analyzed the data, authored or reviewed drafts of the article, and approved the final draft.
- Alemayehu Worku conceived and designed the experiments, analyzed the data, authored or reviewed drafts of the article, and approved the final draft.
- Merga Dheresa conceived and designed the experiments, analyzed the data, authored or reviewed drafts of the article, and approved the final draft.
- Yemane Berhane conceived and designed the experiments, analyzed the data, authored or reviewed drafts of the article, and approved the final draft.

### Human Ethics
The following information was supplied relating to ethical approvals (*i.e.*, approving body and any reference numbers):

The Ethiopian Public Health Institute's Research Ethics Institutional Review Board reviewed and approved the study protocol for implementation (Ref. EPHI 6.13/517).

### Data Availability
The raw data are available in the Supplemental Files.

### Supplemental Information
Supplemental information for this article can be found online at http://dx.doi.org/10.7717/peerj.16144#supplemental-information.

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
