# Peer review of "The effect of drop-in centers on access to HIV testing, case finding, and condom use among female sex workers in Addis Ababa, Ethiopia"

_PeerJ, doi:10.7717/peerj.16144_

## Round 0.1 · original submission · Major Revisions

The reviewers appear to support your submission. They have provided some very objective and insightful comments with respect to points that you must address before this manuscript is ready for publication. Many of these points regard further clarifications with respect to the population and the data and encourage further interpretation of that data. Finally, all of the reviewers strongly recommend additional proofreading and corrections to the grammar.

We look forward to receiving your revised manuscript.

·

Basic reporting

Abdella et al. in this manuscript entitled “The effect of Drop-in centers on access to HIV testing, case finding, and condom use among Female Sex Workers in Addis Ababa, Ethiopia” investigates the impact of drop-in centers offering HIV prevention services in affecting reported lifetime HIV testing, finding HIV positive cases, awareness of serostatus, and consistent condom use among female sex workers (FSWs) in Addis Ababa, Ethiopia. A quasi-experimental study design was used, and respondent-driven sampling was employed to recruit 1,366 FSW participants. Study participants were divided into two groups, those having previously accessed a drop-in center and those never having accessed a drop-in center. Participants were provided with a rapid HIV test using the National HIV testing algorithm and interviewed via a data collection tool uploaded into ODK. A logit regression model was used to analyze the effect of access to drop-in centers on affecting reported lifetime HIV testing, finding HIV positive cases, awareness of serostatus, and consistent condom use among participants. Results showed a significant increase along all four variables for those participants with access to drop-in centers. The authors successfully demonstrated there is evidence to suggest that drop-in centers improved prevention variables including lifetime HIV testing, awareness of status, and condom use.

Basic Reporting
1. The overall structure and readability of the manuscript is good; however, there are numerous minor grammatical issues, typos, and phrasing which should be addressed to increase comprehension. A few examples can be found on the following lines: 63, 67, 69 – 71, 72 -74. The manuscript would benefit from some additional review by a subject matter expert or editor proficient in English.

2. On line 271 – 273 the authors state “Early treatment of HIV facilitates viral suppression to an undetectable limit, making it less likely for someone to transmit the virus during unprotected sex or needle sharing(30)”. The reference provided only addresses the reduced likelihood of transmission through sexual contact, and not needle sharing.

3. Figures are present and relevant; however, would benefit from the aforementioned additional review for consistency with English-language grammar and phrasing. Additional minor readability issue, table 4 uses four acronyms to label the first column.

Experimental design

Materials and Methods
The manuscript features primary research within the aims and scope of the journal. The research question is well-defined, aiming to assess the effect of drop-in centers providing HIV prevention and support services on improving four HIV prevention-related indicators among female sex workers.
1. Use of the terms ‘had access’ and ‘never accessed’/‘no access’ to drop-in centers may be confusing; it may be more appropriate to clarify using something like “have previously accessed” and “never have accessed”. Currently it reads that the authors are saying that some FSW did not have access, however the methods indicate that the single criteria used to determine that was never having used a drop-in center, which shouldn’t be interpreted as not having access.
2. The authors do not address whether seed and/or recruited participants were given an incentive to participate in the study. Due to this, it is not possible to account for whether participants would be influenced by an incentive to falsely disclose behaviors qualifying them for the study.
3. Within the study population sampling procedure, it is not clear how many potential participants were identified via respondent-driven sampling vs. how many (if any) were excluded as ineligible.
4. In the sample size determination, a study seemingly describing the impact of peer intervention on consistent condom use among FSW in Kenya is described, but not cited. It’s not clear how this factored into the sample size determination and needs additional clarification.
5. While describing dependent and independent variables, the term ‘ever tested for HIV’ is used, reporting ‘life-time HIV testing’ may be a more conventional term to describe this variable.
6. On line 130 – 131 the methods describe “The number of seeds selected for each city was based on the estimated total number of FSWs living in Addis Ababa”, however no other study setting is described outside Addis Ababa. This needs additional clarification.
7. Ethical considerations do not address whether newly identified HIV-positive participants (n54) were referred to treatment and support services.

Validity of the findings

Validity of the findings
Findings support previous research showing the benefit provided by access to HIV prevention and support services to FSWs. Findings showed a correlation between previously having accessed a drop-in center and increased likelihood of reporting lifetime HIV testing, awareness of serostatus, and consistent condom use.
1. The authors connect an increased likelihood of finding HIV positive FSWs (7.02%) among those having accessed a drop-in center. However, it is unclear whether participants were questioned on where they received their initial reactive test, and whether they indicated it was through testing at a drop-in center. Additional clarity would be beneficial.
2. A minor issue is it is not described if/how ‘consistent condom use’ was defined for participants, and whether participants were attesting to use of condoms with every act of intercourse.
3. The authors conclude that “it is essential to strengthen the services provided in DICs and expand the centers. This will ensure that the entire network of FSWs is reached with appropriate HIV prevention services”. It would enhance their conclusions to describe what services should be strengthened, and how it might potentially expand access (or engagement) to/among FSWs.

Additional comments

General Comments
The study and manuscript are well structured, employ a moderately well-documented and statistically sound design, and pursues a relevant and meaningful research question with findings with both potential academic and practice-based benefits. Most identified issues with the manuscript revolve around clarity and can likely be addressed through additional revisions by both the authors and colleagues/services proficient in English grammar and phrasing.

Reviewer 2 ·

Basic reporting

This paper aims to assess the effect of the implementation of drop-in centres (DIC) for female sex workers (FSW) in Addis Ababa, Ethiopia, on several HIV outcomes: having ever tested for HIV -and knowing the result), positivity rate, awareness of HIV positive status (for those HIV-positive), and consistent condom use. DIC have been implemented in Addis Ababa since 2016. The authors implemented a respondent-driven sampling (RDS) survey in 2020 and recruited 1366 FSWs. The outcomes were compared between FSWs who reported they had ever accessed to DICs and those who had never accessed DICs. A propensity score matching technique was used to balance the potential confounders. Their results are summarized as: “Access to DIC produced a 7.58% increase in the probability of testing to know HIV status (P < 0.001), a 7.02% increment in ûnding HIV-positive FSWs (P = 0.003), an increase of 6.93% in awareness of HIV status among HIV positive FSWs (P = 0.001), and a 4.39% rise in consistent condom use (P = 0.01).”

A raw dataset is provided. The Stata code of the analysis is not provided.

In the abstract, only the differences are reported, eg “Access to DIC produced a 7.58% increase in the probability of testing to know HIV status”. It would be relevant to add also the two compared proportions to provide a sense of the level of each outcome in the population.

The methods section should be reorganized, taking into account what was already introduced or not to the reader. For example, line 121 “Interviews and blood draws were conducted in a private room”. At that stage, study procedures were not yet introduced.

Several elements of the methods are, in fact, results and should be moved appropriately. For example “There were no dead or unproductive seeds.”

The minimum age to participate was 15 years old. How was consent collected for minors? Was the consent from parents/guardians required? Or did the survey have specific authorizations?

HIV testing was proposed to participants. What was the procedure in case of HIV diagnosis? Any referral or linkage intervention?

The Sample size section is unclear. What is the purpose of this section? To determine the sample size of the survey? I’m lost with the following paragraph: “Peer intervention was found to substantially increase the rate of consistent condom use among 151 Female FSWs in Kenya, leading to an increase from 64.0% to 86.2% 79. This statistic would 152 yield a total sample size of 132.”

Figure 1 is very technical and maybe more appropriate as supplementary material.

Table 1: what is the currency of monthly income? One digit is sufficient for percentages.

Table 2, table 3, and table 4 would benefit to be reformatted to be easier to read by a non statistical reader. For example, use percentages for proportions. What is ATET acronym used in table 4 and not introduced in the paper?

Experimental design

First of all, DICs are not adequately defined in the introduction. Are they HIV clinics providing testing, treatment and biomedical services or just testing services? Are they run by medical staff or by community peers? Are they dedicated to FSWs only? Opened to other key populations? To the general population as well? Could they be considered community-based centres? Or rather community-friendly? There are many models of services in the field of HIV in Africa, and it would be essential to have enough context to characterize these DICs properly.

Somehow, I would be more nuanced about the paper’s main objective. If it is possible to look at the statistical association between having ever visited a DIC and the different outcomes, the methodology does not seem robust enough to determine a causal effect. We do not know when the participants visited a DIC and when they were tested or diagnosed, while consistent condom use was measured over the last 30 days. In the Discussion, I would replace “Access to DICs improves the probability of …” by “Access to DICs was associated with the probability of…”.

The first essential question would be to know who ever visited a DIC and who did not. Some descriptive statistics (using RDS weights) would be welcomed here.

Another limitation is the small number of covariates included in the analysis: age, marital status, educational status, monthly income, and alcohol use. For example, there is nothing about the condition of sex work: number of clients, places of work, age at sex work entry… It would be crucial to have a more comprehensive idea of who was included in the survey and how it is associated with DIC use. The current selection of considered covariates is not explained nor discussed.

Regarding the methodology, the authors used a complex approach with propensity score weighting to recreate, a posteriori, two comparative groups to compare the different outcomes. I’m not a propensity score specialist, and a statistical reviewer’s feedback would be helpful. However, I am unsure about the added value of such a complex approach in this context. Why not use a more conventional multivariate regression? Wouldn’t it be enough to quantify the association between having visited a DIC and the different outcomes? An advantage would be to show the other factors associated with the different outcomes and thus enrich the analysis.

As stated line 260, it is recommended for FSWs to test at least once per year. Would it be relevant to consider recent HIV testing (in the last 12 months) as an additional outcome, particularly considering the current outcome (ever tested) is very high (93%) in this population?

“Finding HIV positive FSWs” outcome. The current naming adopted a provider perspective. Would it be more neutral to consider “positivity rate”? Was HIV testing mandatory to participate in the survey? If not, how were handled refusals to test?

Validity of the findings

As expressed, I would be more nuanced in the Conclusion.

The authors stated "that access to DICs leads to an increase in HIV testing, HIV case finding, HIV status awareness among HIV positives, and consistent condom use by FSWs."

If there is an association between having ever visited a DIC and the outcomes, we cannot properly infer any causal effect.

Additional comments

• Introduction “Evidence has shown that these key populations and their clients comprise two-thirds of global HIV infections”. What was the year of the estimate? Could it be added in the sentence.
• Introduction: “The innovative nature of these centres…” How are they innovative?
• Line 125: “A respondent-driven sampling technique.” A reference is required.
• Line 130: “The number of seeds selected for each city was based on the estimated total number of FSWs living in Addis Ababa.” Missing reference about these estimates.
• § lines 204-215: it is very difficult to follow for a non-specialist. In addition, many acronyms are used without being defined.
• The second sentence of the Discussion is identical to the last sentence of the Results, just few lines before.

·

Basic reporting

Overall the paper could benefit from a proper review of the writing and structuring of sentences to make it more coherent and reduce chances of confusion to the reader. Throughout the paper there are several grammatical errors that need to be reviewed. This includes the tables (e.g. categories for drinking alcohol).
throughout the document the use of "those who accessed DIC" and "those who had access to DIC" are use interchangeable and those mean two different things and affect how the results are viewed
Review labelling of tables: Table 1: Properly label (study participants and not FSW in Ethiopia, add N. Add units (age in years, income in ?currency) Table 3 and 4 could be merged
Results: Line 225 use correct age category 20-29; Line 228 and 230 have 421 and 412 for those who had accessed services in DIC. Please correct. 246-249 essentially those who had accessed DIC had higher probability of having HTS, more likely to be HIV+ and tested +ve at sample collection (new positive). This affects discussion of results
Discussion: There is repeated reference to HIV+ve needing a HTS even when status is already known; row 275, 279.

Experimental design

No comment

Validity of the findings

Row 290: Literature shows that those enrolled in stable care in DIC have low seroconversion rates but in this cohort we see higher new positives in those who have access to DICs. What could possible explain this?

Curious about the general high access to HTS and condoms among even those who did not access DICs in this cohort. Begs the question whether access through community outreaches, health facilities is actually generally higher?

Utility of DICs as important avenue to reach KPs has been well documented. While this paper shows the increase in access to HTS and higher KP PLHIV (need to address other possible reasons why in discussions section e.g. referrals from outreaches etc), need to explain the new HIV positives being higher in this cohort

Additional comments

No comment

---

## Round 0.2 · accepted · Accept

All of the critical points were addressed to make this work suitable for publication. Congratulations.

·

Basic reporting

Basic Reporting
Upon second review, Abdella et al. have addressed many of the areas identified for improvement in the initial review. Grammar and phrasing has noticeably improved in this revised submission, with all example areas being addressed along with a general improvement in readability overall. A reference to TasP was clarified to indicate it was specific to transmission by sexual intercourse. Table 4 remains difficult to read due to the overuse of acronyms in the header.

Experimental design

Experimental Design
The authors rectified a number of issues identified within the previous review that have improved the clarity of the methods and materials. Authors clarified participant eligibility criteria, recruitment methods, setting, and exclusion criteria. Additionally, the authors incorporated standard terminology used withing similar studies that improves clarity overall. Finally, the authors fleshed out additional details on how newly identified reactive cases were referred to treatment, addressing a previous gap in ethical considerations.

Validity of the findings

Validity of the Findings
The authors addressed a number of questions seeking to clarify how study participants responded during data collection, which overall strengthened the author’s interpretation of the findings. Additionally, authors expanded on the practice-based implications of the study, strengthening the relevance of its findings.

Additional comments

Additional Comments
Abdella et al. have significantly improved upon the readability and clarity of the manuscript. It is well-structured, designed, and referenced. The addition of a stronger conclusions section improves upon the applicability of the study.

Reviewer 2 ·

Basic reporting

The authors provided a detailed rebuttal letter and responded to all reviewers' comments.

Experimental design

The authors provided a detailed rebuttal letter and responded to all reviewers' comments.

Validity of the findings

The authors provided a detailed rebuttal letter and responded to all reviewers' comments.

Additional comments

Minor editing is still required. For example, some figures use a comma as a thousand separator, some others do not.